# Formulation and Characterization of Stimuli-Responsive Lecithin-Based Liposome Complexes with Poly(acrylic acid)/Poly(*N*,*N*-dimethylaminoethyl methacrylate) and Pluronic^®^ Copolymers for Controlled Drug Delivery

**DOI:** 10.3390/pharmaceutics14040735

**Published:** 2022-03-29

**Authors:** Mónica G. Simões, Ayelen Hugo, Andrea Gómez-Zavaglia, Pedro N. Simões, Patrícia Alves

**Affiliations:** 1CIEPQPF, Department of Chemical Engineering, University of Coimbra, Rua Sílvio Lima, Pólo II-Pinhal de Marrocos, 3030-790 Coimbra, Portugal; simoesmonica91@gmail.com; 2Center for Research and Development in Food Cryotechnology (CIDCA, CCT-CONICET), La Plata 1900, Argentina; ayelen_h@yahoo.com.ar (A.H.); angoza@qui.uc.pt (A.G.-Z.)

**Keywords:** polymer–liposome complexes, Pluronic^®^-poly(acrylic acid), Pluronic^®^-poly(*N*,*N*-dimethylaminoethyl methacrylate), stimuli-responsive, intelligent drug delivery systems

## Abstract

Polymer–liposome complexes (PLCs) can be efficiently applied for the treatment and/or diagnosis of several types of diseases, such as cancerous, dermatological, neurological, ophthalmic and orthopedic. In this work, temperature-/pH-sensitive PLC-based systems for controlled release were developed and characterized. The selected hydrophilic polymeric setup consists of copolymers of Pluronic^®^-poly(acrylic acid) (PLU-PAA) and Pluronic^®^-poly(*N*,*N*-dimethylaminoethyl methacrylate) (PLU-PD) synthesized by atom transfer radical polymerization (ATRP). The copolymers were incorporated into liposomes formulated from soybean lecithin, with different copolymer/phospholipid ratios (2.5, 5 and 10%). PLCs were characterized by evaluating their particle size, polydispersity, surface charge, capacity of release and encapsulation efficiency. Their cytotoxic potential was assessed by determining the viability of human epithelial cells exposed to them. The results showed that the incorporation of the synthesized copolymers positively contributed to the stabilization of the liposomes. The main accomplishments of this work were the innovative synthesis of PLU-PD and PLU-PAA by ATRP, and the liposome stabilization by their incorporation. The formulated PLCs exhibited relevant characteristics, notably stimuli-responsive attributes upon slight changes in pH and/or temperature, with proven absence of cellular toxicity, which could be of interest for the treatment or diagnosis of all diseases that cause some particular pH/temperature change in the target area.

## 1. Introduction

The so-called smart drug delivery systems aim at targeting effectively a given active agent into a specific location by responding to stimuli such as variations in pH, temperature, light, etc. [1,2]. The role of controlled-release systems in areas such as gene therapy, the treatment and/or diagnosis of cancer, neurological, dermatologic, ophthalmologic and orthopedic diseases, as well as cosmetic products and food engineering, contributes to explaining the hundreds of publications related to this matter [3,4] and also to addressing different challenges of the food and cosmeceutical industries (the targeted release of unstable bioactives, e.g., antioxidants) [5,6]. The importance of these studies came from the continuing need to find more reliable, effective and selective drug release solutions.

Controlled delivery systems can be based on polymeric, inorganic or lipid compounds. Among them, lipids have unique properties that allow the relatively easy formation of nano-sized structures such as liposomes. They are biocompatible, biodegradable, non-immunogenic and non-toxic vesicles, ideal for the encapsulation, transport, storage and release of hydrophilic and/or lipophilic substances [7,8]. In addition, the use of liposomes as carriers enhances the solubility and stability of encapsulated drugs, and reduces their side effects and toxicity [9,10]. However, there are some drawbacks in the in vivo application of these nanocarriers since they are easily attacked and uptaken by phagocytic cells of the immune system. This instability represents a major issue and compromises the efficiency of the drug delivery at the desired target location [2]. To overcome this problem, the anchoring of polymers into the lipid bilayers of liposomes, leading to the so-called polymer–liposome complexes (PLCs), has been a successful strategy [9,11]. The presence of polymers in the liposome surface brings more mechanical resistance and prevents the capture by the phagocyte system, thus increasing the residence time of PLCs in the bloodstream, which is an essential condition for a successful treatment. Poly(ethylene glycol) (PEG)-coated liposomes represent the paradigmatic case of long-circulation liposomes that are commercialized for cancer treatment [12,13,14]. PEGylation involves the grafting of PEG to the surfaces of nanoparticles/liposomes, wherein ethylene glycol units form tight associations with water molecules, resulting in the formation of a hydrating layer [15].

Over the past few decades, PEG has been considered to be non-immunogenic. However, there is growing evidence that it might be more immunogenic than previously recognized. This is supported by the presence of anti-PEG antibodies in healthy humans who are increasingly exposed to PEG additives. Furthermore, there is evidence that formulations containing anticancer drugs in PEGylated liposomes (Doxil^®^, DaunoXome^®^ and Ambisome^®^) could induce CARPA (complement activation-related pseudoallergy), which is classified as a non-IgE-mediated pseudoallergy caused by the activation of the complement system [16]. Therefore, it is very important to find new non-immunogenic, generally recognized as safe (GRAS) polymers capable of extending the half-life of liposomes.

The development of alternative intelligent-release systems based on stimuli-responsive polymers, namely those sensitive to pH and/or temperature, has been rather well accepted in the medicine and pharmaceutical fields [17]. Poly(*N*-substituted acrylamides) are among the most studied thermosensitive polymers [18], and liposomes functionalized with this class of polymers can accurately release the encapsulated drug at temperatures above the lower critical solution temperature [8]. Poly(2-(*N*,*N*-dimethylamino)ethyl methacrylate) (PDMAEMA) has been investigated for gene delivery materials [19], anticancer drug delivery by micelles [20], coating magnetic nanoparticles in cancer treatments [21] and more recently for incorporation into drug delivery liposomes [22]. Poly(acrylic acid) (PAA) is another polymer suitable to deliver and release drugs in tumors and inflammation sites due to its inherent biocompatibility, pH sensitivity and mucoadhesive properties [23,24,25]. PAA has been studied in the form of hydrogels and nanoparticles, and incorporated into liposomes, the latter case corresponding to pH-sensitive PLCs [24,25,26]. Pluronic^®^ (PLU), also known as Poloxamer, is a biocompatible and non-toxic triblock copolymer of polypropylene oxide and ethylene oxide, with known applications in delivery systems [27]. PLU can be incorporated into liposomes to form PLCs due to the hydrophobic nature of polypropylene oxide, which promotes the polymer anchoring into the lipid bilayer [28,29,30]. It has been reported that PLU increases the permeation of a drug through the blood–brain barrier, affects the micro-viscosity of cells and is also capable of sensitizing and accumulating in multidrug-resistant cancer cells [31,32].

In this work, we are proposing a novel step that goes beyond previous attempts (e.g., [33]) towards the conjugation of these polymers. Here, we formulate PLCs that combine all benefits of both polymers and liposomes in just one control release system. Therefore, we are employing PLU, as a nuclear element, conjugated with PAA and PDMAEMA segments, as stimuli-responsive copolymers that can be used to formulate long-circulation pH-/temperature-sensitive PLCs. Moreover, the synthetized PLU-PAA and PLU-PD are hydrophilic polymers that could form three-dimensional networks capable of holding a large amount of water. It is possible that these polymers could also form a hydrating layer in our liposomes, as PEG does, protecting them from protein adsorption and the subsequent opsonization and destabilization.

PLU was used as an initiator in the PDMAEMA/PAA polymerization reaction, after esterification with bromide 2-bromoisobutyryl (2-BiB) [22]. The copolymers were synthesized under mild reaction conditions by control/living radical polymerization (LRP), particularly by atom transfer radical polymerization (ATRP) [34,35]. This technique allows one to obtain low molecular weights and low dispersity (Ð), which are crucial parameters to achieve an efficient polymer-based drug delivery system.

Liposomes were formulated with soybean lecithin (LC), which is a non-toxic, natural phospholipid found in the organism and is also used as a food supplement [36,37]. Given the intended drug delivery applications, release profiles, calcein loading capacity (CLC), stability at different pH/temperatures and cell viability were evaluated to select the most appropriate copolymer and copolymer/lipid ratio for the proposed PLC systems.

## 2. Materials and Methods

The following materials were used in the different stages of polymeric synthesis and PLC formulation: 1,1,4,7,7-pentamethyldiethylenetriamine (98%, Alfa Aesar, Kandel, Germany), 3-(4,5-dimethylthiazol-2-yl)-2,5-diphenyltetrazolium bromide (MTT, Sigma-Aldrich, St. Louis, MO, USA), 4-(2-hydroxyethyl)-1-piperazineethanesulfonic acid (HEPES, Amresco^®^, Seattle, WA, USA), 4-(dimethylamino)pyridine (PMDETA, 99%, Merck, Darmstadt, Germany), bromide 2-bromoisobutyryl (2-BiB, 97%, Alfa Aesar, Kandel, Germany), calcein (Acros Organics, Geel, Belgium), chloroform (99.2%, VWR Chemicals, Fontenay-sous-Bois, France), copper (I) bromide (99%, Alfa Aesar, Kandel, Germany), deuterated dimethyl sulfoxide (99.9%, Sigma-Aldrich, St. Louis, MO, USA), dialysis membranes (MWCO 3500 Da, Medicell Membranes Ltd., London, UK), dichloromethane (99.8%, VWR Chemicals, Fontenay-sous-Bois, France), dimethyl sulfoxide (DMSO, 99.6%, VWR Chemicals, Fontenay-sous-Bois, France), Dulbecco’s Modified Eagle Medium (GIBCO BRL Life Technologies, Rockville, MD, USA), fetal bovine serum (FBS, PAA Laboratories, GmbH, Pasching, Austria), hydrochloric acid (37%, VWR Chemicals, Fontenay-sous-Bois, France), methanol (99%, VWR Chemicals, Fontenay-sous-Bois, France), *N*,*N*-dimethylaminoethyl methacrylate (DMAEMA, ≥99%, Merck, Darmstadt, Germany), n-hexane (99.3%, VWR Chemicals, Fontenay-sous-Bois, France), non-essential amino acids (GIBCO BRL Life Technologies, Rockville, MD, USA), penicillin–streptomycin solution (GIBCO BRL Life Technologies, Rockville, MD, USA), Pluronic^®^ F68 (Sigma-Aldrich, St. Louis, MO, USA), sodium chloride (99%, Sigma-Aldrich, St. Louis, MO, USA), soybean lecithin (LC, Acros Organics, Fair Lawn, NJ, USA), tert-butyl acrylate (tBA, 99%, Alfa Aesar, Kandel, Germany), tetrahydrofuran (VWR Chemicals, Fontenay-sous-Bois, France), toluene (99%, VWR Chemicals, Fontenay-sous-Bois, France), triethylamine (99%, Merck, Darmstadt, Germany), trifluoracetic acid (99%, VWR Chemicals, Fontenay-sous-Bois, France) and Triton^®^ X-100 (Sigma-Aldrich, St. Louis, MO, USA).

### 2.1. Synthesis of Pluronic^®^-2-Bromoisobutyrate (PLU-Br)

The initiator was obtained by the esterification of Pluronic^®^ F68 (PLU) with bromide 2-bromoisobutyryl (2-BiB) (Figure 1A). This approach is based on the synthesis of cholesterol-2-bromoisobutyrate (CHO-Br) reported by Alves et al. [18]. Briefly, a 1 g sample of 4-(dimethylamino)pyridine (previously recrystallized from toluene) in 10 mL of dry dichloromethane was mixed with 0.7 mL of triethylamine (dried over CaH_2_ and vacuum-distilled). The solution was transferred to a 250 mL three-neck round-bottom flask equipped with a condenser, dropping funnel, gas inlet/outlet and a magnetic stirrer. After cooling to 0 °C, 1.5 mL of 2-BiB in 10 mL of dry dichloromethane was added. Then, 21 g of PLU in 50 mL of dry dichloromethane was added dropwise to the formed yellow dispersion, for 1 h under dry nitrogen; subsequently, the temperature was raised to 28 °C. The reaction was kept under stirring for 22 h. Afterwards, the mixture was washed with a saturated aqueous sodium chloride solution and dried over magnesium sulfate, followed by evaporation of half of the solvent. The PLU-Br initiator was precipitated in ethanol and finally filtered and dried in vacuum. The final product was obtained in the form of a white powder and characterized by ^1^H NMR.

### 2.2. Synthesis of Pluronic^®^-Poly(N,N-dimethylaminoethyl methacrylate) (PLU-PD)

PLU-PD was synthesized by ATRP (Figure 1B) according to Eugene et al. [33]. To obtain the copolymer, the previously prepared PLU-Br was used as an initiator, copper (I) bromide as a catalyst, 1,1,4,7,7-pentamethyldiethylenetriamine (PMDETA) as a ligand and toluene as a solvent. Succinctly, the monomer *N*,*N*-dimethylaminoethyl methacrylate (DMAEMA, 2 mL freshly passed through a Al_2_O_3_ column) and 600 mg of the initiator PLU-Br were added to a 25 mL Schlenk flask equipped with a magnetic stirrer, and frozen and bubbled with N_2_ to eliminate oxygen. Then, 30 mg of copper (I) bromide, 26 mg of PMDETA and 2 mL of toluene (previously bubbled with N_2_) were added under N_2_ atmosphere to the Schlenk flask, which was then sealed and deoxygenated under reduced pressure and then filled with N_2_. The reaction proceeded for 24 h at 80 °C, after which it was quenched by the addition of acetone and exposure to air. Purification was achieved by passing the obtained product through a neutral alumina column using acetone as the eluent, then dialyzed for 48 h, dried by evaporation, dissolved in water, frozen at −20 °C for 24 h and freeze-dried at −50 °C and 0.04 mbar in an Alpha 1–2 LD Plus (CHRIST, Osterode am Harz, Germany) for an additional 48 h. The obtained polymer was analyzed by gel permeation chromatography (GPC) and nuclear magnetic resonance (^1^H NMR).

### 2.3. Synthesis of Pluronic^®^-Poly(tert-butyl acrylate) (PLU-PtBA)

PLU-PtBA was also obtained via ATRP (Figure 1C) from the same procedure used in the synthesis of PLU-PD previously described, except for the monomer used, which was tert-butyl acrylate (tBA) instead of DMAEMA; also different were the reagent quantities (500 mg of PLU-Br, 24 mg of copper (I) bromide and 30 mg of PMDETA) and reaction time, which, in this case, was 30 min. Once purified and isolated, the obtained polymer was analyzed by GPC and ^1^H NMR.

### 2.4. Synthesis of Pluronic^®^-Poly(poly(acrylic acid) (PLU-PAA)—PLU-PtBA Hydrolysis

PLU-PtBA was hydrolyzed in order to obtain PLU-PAA (Figure 1D) [38]. Briefly, PLU-PtBA was added to a round-bottom flask with three necks (equipped with a condenser, an addition funnel and magnetic stirring) with 10 mL of dichloromethane, in a bath at 0 °C under N_2_ atmosphere. A 5-fold molar excess of trifluoroacetic acid was added dropwise to the flask, and then the temperature was slowly raised to 30 °C. After 48 h of reaction, the polymer was precipitated in n-hexane, filtered and dried under vacuum at 40 °C until a constant weight. PLU-PAA was characterized by ^1^H NMR.

### 2.5. Characterization of the Initiator and Polymers

#### 2.5.1. Nuclear Magnetic Resonance (NMR)

^1^H NMR spectra were collected in a Bruker Avance III 400 MHz spectrometer (Bruker, Billerica, MA, USA), by using deuterated dimethyl sulfoxide as a solvent and tetramethylsilane (TMS) as an internal standard, in 5-mm-diameter tubes.

#### 2.5.2. Gel Permeation Chromatography (GPC)

The number-average molecular weight (M_n_) and dispersity (Ð = M_w_/M_n_) of PLU-PtBA and PLU-PD were determined by GPC, in a Viscotek (Dual detector 270, Viscotek, Houston, TX, USA), with THF as the eluent at 30 °C (1.0 mL/min). Narrow polystyrene standards were used for the calibration. OmniSEC software (Malvern Instruments, Malvern, UK) was used along with the TriSEC calibration to determine the M_n,GPC_ and Ð of the obtained polymers.

### 2.6. Preparation of Liposomes and Polymer–Liposome Complexes (PLCs)

Liposomes were prepared according to the hydration film method [3,22,24], which consists of dissolving soybean lecithin (LC) in chloroform (2.9 mM) in a round-bottom flask. A N_2_ stream was then used to evaporate all the chloroform, thus allowing the formation of a thin lipid film. Bare liposomes (LIP) were obtained by rehydrating the lipidic film in a buffered solution of 50 mM 4-(2-hydroxyethyl)-1-piperazineethanesulfonic acid (HEPES) at pH 7.0, vigorously stirred and incubated for 24 h above the transition temperature (ca. 37 °C). PLCs were obtained by adding the copolymer solution of PLU, PLU-PD or PLU-PAA in HEPES (10 mg/mL) to the LIP in copolymer/LC molar ratios of 2.5, 5 and 10%. The formulations were further vigorously stirred and left for a further 24 h at 37 °C for the incorporation of polymers into the liposome bilayers, leading to PLCs.

### 2.7. Particle Size and ζ-Potential Measurements

A Malvern Instrument Zetasizer Nano-Z (Malvern Instruments, Malvern, UK) was used to evaluate particle size and ζ-potential measurements, at 37 °C. The average hydrodynamic particle size (Z-average) and polydispersive index (PdI) were determined by dynamic light scattering at backward scattering (173°) with the Zetasizer 6.20 (Malvern Instruments, Malvern, UK). ζ-potential was determined using a combination of measurement techniques: electrophoresis and laser Doppler velocimetry (Laser Doppler Electrophoresis) (Malvern Instrument Zetasizer Nano-Z, Malvern Instruments, Malvern, UK). ζ-potential outcomes were provided directly by the instrument. The results presented are the average and standard deviation of at least 10 replicates per sample.

### 2.8. Retention Capacity and Leakage Experiments

Calcein was used as a fluorescent dye for the release experiments by means of its encapsulation into the aqueous compartment of the liposomes. Briefly, the lipid films obtained after chloroform evaporation were hydrated with 60 mM calcein in HEPES (50 mM, pH 7.0), vigorously stirred and incubated for 24 h above the transition temperature (ca. 37 °C) to obtain calcein-loaded LIPs. The incorporation of the copolymers into calcein-loaded PLCs followed the procedure previously described. To eliminate the non-entrapped calcein from the external medium, LIPs and PLCs were centrifuged (10,000 rpm for 5 min, three times; Centurion Scientific Ltd., Stoughton, UK). Washed PLCs were resuspended in 200 µL of HEPES (50 mM, pH 7.0) [22,39].

Release profiles and drug retention capacity along time (up to 35 h) of all formulations were determined fluorometrically on a Synergy HT fluorescence microplate reader (Bio-Tek Instruments, Winooski, VT, USA) with excitation and emission wavelengths at 485/20 nm and 528/20 nm, respectively.

Calcein-loaded LIPs and PLCs were used to evaluate the stability of the liposomes after 15 min incubation time under physiological conditions (37 °C, pH 7.0), and at pH 2, 4, 7 and 11 (37 °C), and at 42 °C (pH 7.0). The fluorescence intensity was determined along time. After the last recording, to induce its total lysis (100% release of calcein), 20 µL of a solution of Triton X-100 (10%, *v*/*v*) was added to each sample [22]. Calcein release percentage was obtained according to Equation (1).
(1)$$\%\;Release=\frac{\left(F-F_{i}\right)}{\left(F_{t}-F_{i}\right)}\times 100$$
where (*F*) is the fluorescence intensity of the sample after each incubation time, (*F_i_*) is the initial fluorescence intensity of the sample, and (*F_t_*) is the total fluorescence intensity of the sample after the addition of Triton-X100. The calcein loading capacity, *CLC* (%), of each test was determined as the molar concentration of calcein per molar concentration of lipid (Equation (2)).
(2)$$CLC\left(\%\right)=\frac{\left[Calcein\right]}{\left[Lipid\right]}\times 100\;$$

A calibration curve ([*Calcein*] = 1 × 10^−3^ (*F_t_ − F_i_*), R^2^ = 0.99) was used to obtain the molar concentration of the encapsulated calcein. The molar concentrations of lipids in LIPs and PLCs were obtained by using a commercial kit (CHO-POD enzymatic colorimetric from Spinreact, Lisbon, Portugal) following the instructions of the manufacturer, and the concentrations were normalized to the total lipid content [40,41].

### 2.9. Cytotoxicity Assays

Cell viability was determined by assessing mitochondrial dehydrogenase activity, using 3-(4,5-dimethylthiazol-2-yl)-2,5-diphenyltetrazolium bromide (MTT). For use in the assays, human epithelial HEp-2 cells (ATCC, Manassas, VA, USA) were cultured in DMEM (Gibco, Grand Island, NY, USA) supplemented with 10% (*v*/*v*) heat-inactivated (30 min/60 °C) fetal bovine serum, and 1% (*w*/*v*) non-essential amino acids and 1% (*v*/*v*) penicillin–streptomycin solution (100 U/mL penicillin G, 100 µg/mL streptomycin). HEp-2 cells were seeded in 48-well plates at 1 × 10^5^ cells per well and incubated at 37 °C in a 5% CO_2_ 95% air atmosphere, to early post-confluence.

LIP and PLCs (75, 375 and 750 µM, in both cases) were added to the cells, in triplicate, and incubated for 24 h. The cells were then washed twice with PBS and the medium replaced with DMEM (without phenol red dye) containing 0.5 mg/mL MTT. After 2 h incubation, 0.2 mL dimethylsulfoxide was added to each well and stirred for 20 min at 25 °C on a plate shaker to solubilize the cells and the formed formazan crystals. The optical density (OD) values were collected in a Synergy HT fluorescence microplate reader at 490 nm (Bio-Tek Instruments, Winooski, VT, USA). *Cell viability* was calculated by Equation (3).
(3)$$Cell\;Viability\;\left(\%\right)=\frac{{\mathrm{OD}}_{t}}{{\mathrm{OD}}_{c}}\times 100\%$$
where OD_t_ is the optical density of the cells treated with liposomes, and OD_c_ is the optical density of the non-treated control cells.

## 3. Results

The synthesis success of the two copolymers was confirmed by ^1^H NMR (Figure 2). From PLU (Figure 2A) to PLU-Br spectra (Figure 2B), a peak at 2.1 ppm appears (d, CH_3_C–Br), indicating the presence of the 2-bromo-2-methylpropionyl groups from the esterification of the PLU with the 2-BiB, confirming the modification/initiator synthesis [22,33]. Figure 2C shows the PLU-PtBA spectrum with a new peak at 1.45 ppm. This chemical shift is ascribed to the methyl protons (e, −C(CH_3_)) of the tBA segments. PLU-PAA (Figure 2D) was obtained by removing the tert-butyl groups of PLU-PtBA by acidic hydrolysis, as shown in Figure 2C. In Figure 2D, after the hydrolysis, the chemical shift of tert-butyl groups (1.45 ppm) vanishes completely, which confirms its success. In Figure 2E, the signals at 2.2–2.4 ppm are ascribed to methyl (f, N–CH_3_) and methylene (g, N–CH_2_) protons of DMAEMA segments, and at 4.2 ppm to the methylene protons adjacent to the oxygen moieties of the ester linkages (h, H_2_CO–C=O). The number-average molecular weight (M_n_) and dispersity (Đ = M_w_/M_n_) of PLU-PtBA and PLU-PD obtained from GPC analysis are included in Figure 1. The success of the polymerization approach is supported by both Ð close to one and the high conversion level (99% and 84% for PLU-PD and PLU-PAA, respectively), the latter obtained by peak integration of the ^1^H NMR spectra.

LIPs’ and PLCs’ particle size and ζ-potential were assessed at 37 °C and pH 7.0. PLCs were prepared with copolymer/lipid ratios of 2.5, 5 and 10%, and the corresponding Z-average, PdI and ζ-potential are gathered in Table 1. Comparing LIPs with PLCs, an increase in particle size is observed after the functionalization. Generally, PLCs’ Z-average increases with the percentage of copolymer added. On the other hand, PdI changes slightly, although it is below 0.5 in every case. Vesicles with a higher percentage of PLU-PD exhibit lower absolute values of ζ-potential. The opposite effect can be noticed in PLU-PAA PLCs, while LIPs are the most negative vesicles studied. The CLC results are also summarized in Table 1. PLCs present higher EE than LIP for all copolymer/lipid ratios, and the PLCs with 5% of PLU-PD/PLU-PAA are those with the highest CLC (ca. 27%).

PLCs’ stability was estimated by measuring calcein release at 37 °C and pH 7 (Figure 3). All samples exhibit lower releases when compared to LIPs. PLCs with 10% of PLU-PD/PLU-PAA show a release of ca. 25% after 30 h, which contrasts with LIPs, which release around 60% of their encapsulated calcein after the same time. Moreover, the increase in the PLCs’ copolymer/lipid ratio also lowers the release rate, which indicates enhanced efficiency in retaining the content. Among copolymers, PLU-PD was demonstrated to be more effective than PLU-PAA, considering that, for the same incubation time and amount of copolymer, the PLU-PAA PLCs are faster releasers (i.e., 5% PLCs, Figure 3).

Calcein release was also determined at different pHs (from 2 to 11, at 37 °C) and different temperatures (37 and 42 °C, at pH 7), as shown in Figure 4 and Figure 5a, respectively. When slight changes in pH were applied (either higher or lower then 7.0), PLCs became significantly less stable (Figure 4). As expected, the same outcome was not observed in LIPs (Figure 4). Among the copolymers, the pH destabilization is more evident with PLU-PAA PLCs (i.e., 10% PLCs, Figure 4). On the other hand, PLU-PD PLCs exhibit a faster and higher calcein release at 42 °C (ca. 22%) for the same incubation time (Figure 5a).

Regarding cell viability studies, Figure 5b depicts the percentage survival of HEp-2 cells when exposed to different concentrations of LIPs and 10% PLU-PD/PLU-PAA PLCs, after 24 h. The results demonstrate that, for the tested concentrations, LIPs and/or PLCs are revealed to be non-cytotoxic to HEp-2 cells.

## 4. Discussion

As far as we are aware, the synthesis of PAA or PDMAEMA copolymers using the ATRP method with PLU-Br as an initiator has never been reported in the context of PLC formulation. Compared with other LRP methods, e.g., oxyanion-initiated polymerization [35,42] and reversible addition-fragmentation chain transfer (RAFT) polymerization [43], ATRP (Figure 1 and Figure 2) is simpler and avoids severe conditions. Moreover, the LRP approach stands out due to the strict control of the copolymer homogeneity in terms of both structure and molecular weight, allowing low Ð values and high conversion levels to be achieved. The advantage of using PLU-PD and PLU-PAA copolymers for the formulation of PLCs rests on the PLU hydrophobic segment and its high affinity with the lipid membranes, which enables and facilitates their incorporation [44]. Furthermore, thanks to the stimuli-responsive character of PAA and PDMAEMA segments [22,24], pH-/temperature-sensitive PLCs could be developed.

The low Z-average and PdI results (Table 1) reveal acceptable homogeneity and size distributions of the formulated PLCs, which are key parameters for several biomedical applications. For instance, small particles (e.g., liposomes) are able to accumulate in tumors due to their higher permeation capacity [3,4,45]. Nevertheless, further studies are necessary in order to tune these parameters according to the targeted application.

Concerning the ζ-potential, the observed trends can be explained by the positive and negative character of PDMAEMA and PAA, respectively (Table 1). The PDMAEMA and PAA segments of the synthesized copolymers, PLU-PD and PLU-PAA, are located at the external side of the liposomes. Therefore, the surface charge of PLCs is determined by the polymer and tuned according to the copolymer/lipid ratios (Table 1). In addition, the ζ-potential values are significantly negative, which contributes to preventing the agglomeration of the developed PLCs, and thus to keeping them stable over time, as desired. Taking into account both surface charge distribution and copolymer size, the stabilization of electrosteric quality is plausible [46].

The stabilizing effect of PLU-PD and PLU-PAA was confirmed by the PLCs’ improved CLC (Table 1) and low release profiles obtained at 37 °C and pH 7.0, when compared with LIPs (Figure 3). The copolymers incorporated on the LIP surface avoid cell uptake and act as an extra barrier to the encapsulated dye, preventing its escape. Even if, during the PLC washing, some calcein is lost [3,9,47], a larger amount of dye is kept in the inner aqueous compartment of the liposomes when conjugated with the copolymers, thus explaining the significant difference between LIPs’ and PLCs’ CLC (Table 1).

In a simulated physiological medium for release (37 °C and pH 7.0), LIPs and PLCs show again a distinct behavior (Figure 3). LIPs, which present a low CLC (ca. 11%), are also less stable than any PLC formulation, releasing up to 60% of their content after 30 h (Figure 3). On the contrary, PLU-PD and PLU-PAA PLCs show lower release profiles. Moreover, increasing the copolymer/lipid ratio in the complex, the release becomes slower, ca. 25% after 30 h for 10% PLU-PDMAEA/PAA (Figure 3), which indicates that, from both formulations, the PLCs’ content release can be controlled and tuned.

Additionally, in acid and alkaline medium, at 37 °C, PLCs show pH stimuli-responsive properties in the form of a sharp and high percentage release profile at certain pH values (Figure 4). A possible explanation for this behavior lies in the conformational changes of PDMAEMA/PAA when exposed to mild to acidic or alkali conditions (Figure 4) [22,24]. The fast release at low pH is especially relevant for cancer treatments, because tumor areas are typically acidic, which can promote the drug release only in that specific zone [48]. The PLCs’ destabilization above pH 7 may appear less interesting, although, for other purposes, such as diagnosis or therapeutic monitoring, it might be a relevant attribute.

These results confirm the development of pH stimuli-responsive PLCs both with PLU-PD and PLU-PAA, which are possible candidates for pH-sensitive drug delivery systems [35,42,49].

Contrariwise, for the thermal stimuli experiments at 42 °C and pH 7, the difference between PLU-PD and PLU-PAA PLCs is notorious, as shown in Figure 5a. LIP and PLU-PAA PLCs released a low percentage of calcein after 2 h of incubation (ca. 10%), comparable to the release profiles observed at 37 °C (Figure 3 and Figure 4), indicating no sensitivity to the temperature change. In contrast, PLU-PD PLCs released ~23% of the total encapsulated calcein in the same 2 h, which is a result substantially different from that observed at 37 °C (ca. 5% after 2 h and ~25% after 30 h of incubation) (Figure 3 and Figure 4), revealing a clear sensitivity to higher temperatures [43]. This characteristic of PLU-PD PLCs is interesting, particularly for cases where the desired release site presents higher temperatures, above 37 °C as, e.g., in tumors and adjacent regions or local infections [22].

Cell viability studies are essential in the development of appropriate PLC formulations. The results presented in Figure 5b demonstrate that all studied PLCs reveal no cytotoxicity towards HEp-2 cells within the range of the tested concentrations. The percentages above 100% indicate that all cells were able not only to survive but also to reproduce in the presence of the LIP and PLCs (Figure 5b). Therefore, all the results of this work consolidate the potential of the developed PLU-PDMAEMA and PLU-PAA PLCs as long-circulation stimuli-responsive drug delivery systems for biomedical applications.

## 5. Conclusions

PLU-PD and PLU-PAA copolymers were successfully synthesized by ATRP with PLU-Br as the initiator, the latter obtained by esterification of PLU with 2-BiB. Low-molecular-weight and -dispersity copolymers were obtained, which is a crucial feature towards the formulation of PLCs.

The incorporation and anchoring of these copolymers in liposomes resulted in long-term functional PLCs with acceptable average size and polydispersity and without a tendency to aggregate. The achieved low and controlled release, considerable retention (CLC) and pH/temperature sensitivity are distinctive factors given that temperatures above physiologic levels and low pH are common in tumors and inflammation regions. Furthermore, no cytotoxicity towards HEp-2 cells was detected.

Overall, safe and stable PLCs with pH/thermal stimuli-responsive properties were developed, which can be loaded with any type of hydrophilic or hydrophobic active compound for the treatment and/or diagnosis of a large number of diseases.

## Figures and Tables

**Figure 1 pharmaceutics-14-00735-f001:**
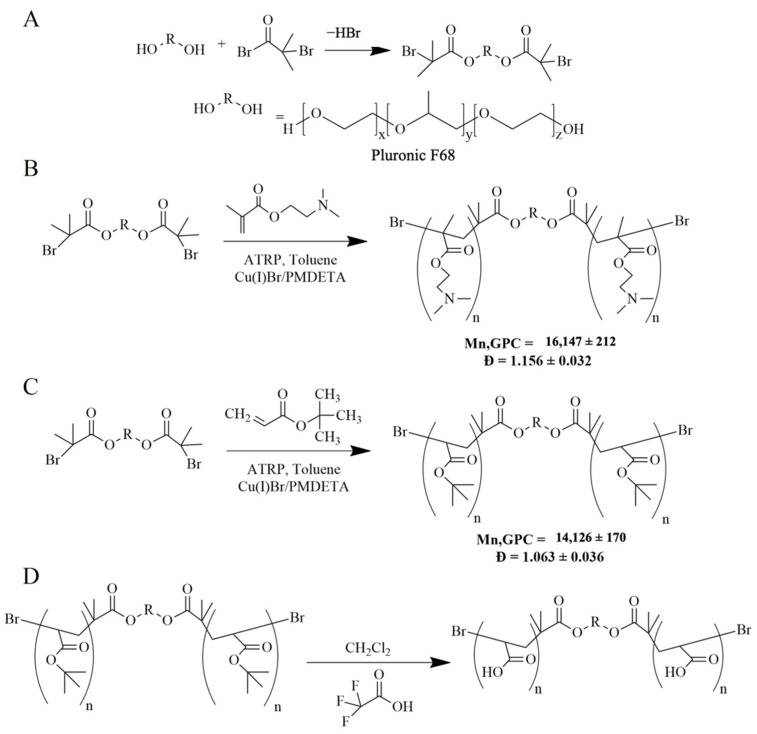
Schematic representation of the (**A**) synthesis of Pluronic^®^-2-bromoisobutyrate (PLU-Br), (**B**) synthesis of Pluronic^®^-poly(*N*,*N*-dimethylaminoethyl methacrylate) (PLU-PD), (**C**) synthesis of Pluronic^®^-poly(tert-butyl acrylate) (PLU-PtBA) and (**D**) synthesis of Pluronic^®^-poly(acrylic acid) (PLU-PAA). Molecular weight and dispersity obtained by GPC analysis of PLU-PD and PLU-PtBA are also presented.

**Figure 2 pharmaceutics-14-00735-f002:**
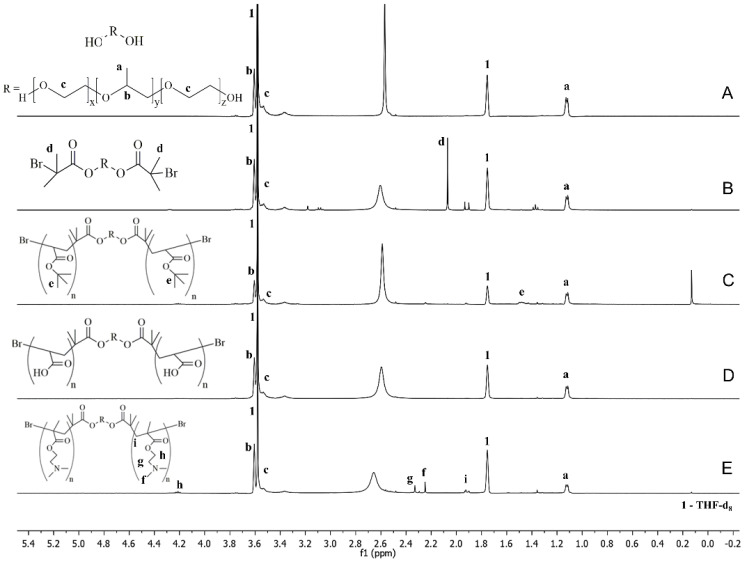
^1^H-NMR spectra of (**A**) Pluronic^®^ F68, (**B**) PLU-Br, (**C**) PLU-PtBA, (**D**) PLU-PAA and (**E**) PLU-PD, where (a) –CH_3_ is the methyl protons of PPO units; (b) –CH is the methine proton of PPO units; (c) –CH_2_ is the methylene proton of PEO units; (d) CH_3_C–Br, from 2-bromo-2-methylpropionyl group; (e) –C(CH_3_) is the methyl protons of tBA units; (f) N–CH_3_ is the methyl protons of DMAEMA units; (g) N–CH_2_ is the methylene protons of DMAEMA units; (h) H_2_C–O–C=O is the methylene protons of DMAEMA units; (i) −CH_2_− from PDMAEMA backbone.

**Figure 3 pharmaceutics-14-00735-f003:**
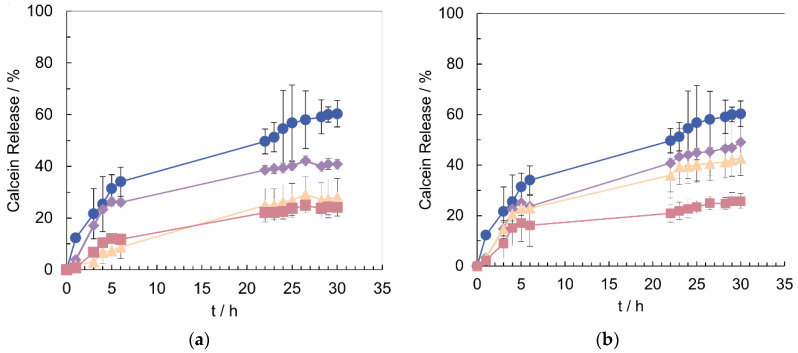
Calcein release profiles at 37 °C and pH = 7. Blue circles: LIP; purple diamonds: 2.5% of copolymer; orange triangles: 5% of copolymer; pink squares: 10% of copolymer. (**a**) PLU-PD PLCs. (**b**) PLU-PAA PLCs. Results are expressed as mean ± standard deviation (SD), n = 3.

**Figure 4 pharmaceutics-14-00735-f004:**
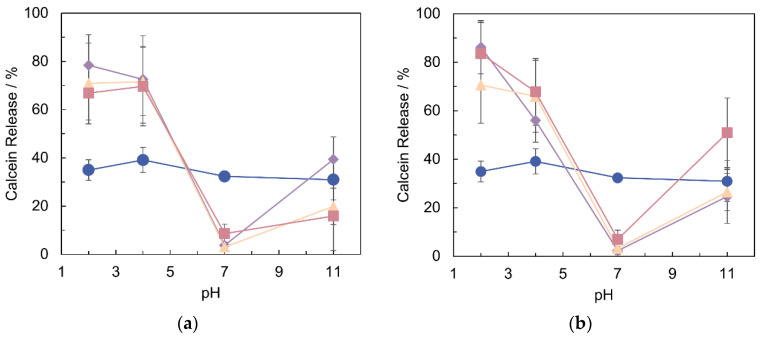
Calcein release profiles after 2 h of incubation at 37 °C and different pHs. Blue circles: LIP; purple diamonds: 2.5% of copolymer; orange triangles: 5% of copolymer; pink squares: 10% of copolymer. (**a**) PLU-PD PLCs. (**b**) PLU-PAA PLCs. Results are expressed as mean ± standard deviation (SD), n = 3.

**Figure 5 pharmaceutics-14-00735-f005:**
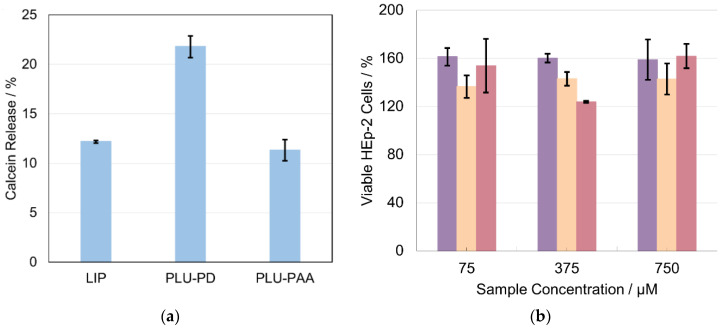
(**a**) Calcein release of LIP, PLCs with 10% PLU-PD and PLCs with 10% PLU-PAA after 2 h of incubation at 42 °C and pH = 7. Results are expressed as mean ± standard deviation (SD), n = 3. (**b**) Viability of HEp-2 cells after 24 h of incubation with LIP (purple bars), PLCs of 10% PLU-PD (orange bars) and PLCs of 10% PLU-PAA (pink bars). Results are expressed as mean ± standard deviation (SD), n = 3.

**Table 1 pharmaceutics-14-00735-t001:** Physical characterization of LIPs and PLCs at 37 °C and pH = 7. Molar concentration of encapsulated calcein, molar concentration of lipids and CLC. Data denoted as mean ± standard deviation (SD), n = 5.

Formulation	Z-Average (nm)	PdI	ζ-Potential (mV)	[Calcein] (mM)	[Lipid] (mM)	CLC (%)
LIP	236.1 ± 7.1	0.439 ± 0.06	−28.2 ± 4.6	0.029 ± 0.002	0.259 ± 0.04	11.1 ± 0.4
PLU-PD 2.5%	267.7 ± 8.9	0.377 ± 0.05	−24.5 ± 6.4	0.064 ± 0.004	0.256 ± 0.05	25.1 ± 1.5
PLU-PD 5%	351.0 ± 8.0	0.433 ± 0.03	−21.0 ± 3.2	0.074 ± 0.005	0.269 ± 0.02	27.3 ± 1.7
PLU-PD 10%	362.5 ± 7.7	0.426 ± 0.04	−18.9 ± 8.2	0.054 ± 0.006	0.246 ± 0.07	21.9 ± 2.5
PLU-PAA 2.5%	244.9 ± 8.3	0.484 ± 0.08	−20.3 ± 2.5	0.041 ± 0.002	0.210 ± 0.03	19.6 ± 0.8
PLU-PAA 5%	225.1 ± 2.8	0.436 ± 0.07	−24.8 ± 3.1	0.064 ± 0.006	0.236 ± 0.03	27.0 ± 2.7
PLU-PAA 10%	253.7 ± 7.2	0.397 ± 0.08	−26.2 ± 8.8	0.041 ± 0.002	0.226 ± 0.05	18.1 ± 0.9

## Data Availability

The raw/processed data required to reproduce these findings cannot be shared at this time due to technical or time limitations, but will be sent upon request.

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
