# Peer review of "Formulation and Characterization of Stimuli-Responsive Lecithin-Based Liposome Complexes with Poly(acrylic acid)/Poly(N,N-dimethylaminoethyl methacrylate) and Pluronic® Copolymers for Controlled Drug Delivery"

_pharmaceutics, 2022, doi:10.3390/pharmaceutics14040735_

Round 1
Reviewer 1 Report
The study of Simões et al regards an interesting topic, the one of developing new types of drug carriers, based on polymer-liposome complexes (PLCs). They investigated two types of PLC systems: temperature or pH sensitive: copolymers of Pluronic F68 - poly(acrylic acid) (PLU-PAA) and Pluronic F68 - poly(N,N-dimethylaminoethyl methacrylate) (PLU-PD). The authors made laborious syntheses, showing a special care to purify the obtained products, they properly characterized the new synthesized systems from the material point of view and clearly presented their properties. In future work, more advanced biological tests will have to be performed to be able to argue that: “Overall, safe and stable long-circulation PLCs …… were developed”. The article is well written and correctly structured. In my opinion, it will attract the interest of many researchers in the field of materials with medical applications.
The English looks very good. I have just a small suggestion:
Line 286 - “…..PLU-PD/PLU-PAA are those with highest the EE…….” Should be replaced with “…..PLU-PD/PLU-PAA are those with the highest EE…….”.
Reviewer 2 Report
In this work authors described the formulation of polymer liposomes complexes (PCLs) by using PDMAEMA and PAA in combination with Pluronics copolymers. they used a specific technique (atom transfer radical polymerization (ATRP)) for a controlled polymerization of the single unit. They produced different PLCs with different ratios between lipids and polymers. they perfomed characterization of nanoparticles testing also their cytotoxicity and their ability to respond to pH/ temperature stimuli in vitro. My major concern regards the diamension and PDI of formulated nanoaprticles. They are really big, while recomended diameter should range between 80 and 200 nm (PDI should be 0,1-0,2). These values indicate the presence of different population of nanoparticles with different diameters. Do the authors tryed to prepared nanoparticles with other appraches? A sonication step or a filtration of nanoparticles solution after polymer inclusion could omogenized the populations. Please clarify this point.
Minor concerns:
-please highlight somehow the different paragraph in materials and methods to have a better view of the various steps
-Figure 2 In the last NMR spectra there is the letter i instead of h as indicated in the caption. Please clarify this point
-line 286 Please correct "the"
-sometimes names in the text and names in figure's labels are different. For example in figure 3,4 and 5 there are PPD and PP labels in the graphs, what do they state for? This could lead to misunderstanding. Please make them uniform.
-line 321 please correct the figure name
Reviewer 3 Report
The manuscript by Simões and coworkers describes the development and characterization of temperature/pH-sensitive polymer-liposome complexes-based systems for controlled release. The manuscript is well structured. However, a series of recommendations are presented below prior publication in Pharmaceutic:
Page 1, line 23: Please “replace eukaryotic cells” by “human epithelial cells” since the PLCs were not tested on other types of eukaryotic cells.
Page 1, lines 36-37: This is not properly the definition of a smart release system. The authors should develop the concept. Indeed, the definition given is more related to a targeting system.
Page 1, line 39: How do the control release systems have a role in cosmetics products and food engineering? The authors should briefly develop this aspect.
Page 2: Check English (grammatical structures and tenses) throughout the introduction.
Page 2, lines 55-59: It would be relevant to emphasize that to the date no other polymer has exhibited better performance for increasing the circulation time of liposomes, but a repetitive administration may lead to the CARPA effect.
Page 2: I would suggest synthesizing the second paragraph (lines 43-59) and the third paragraph (lines 60-79). I believe the research background provided is somewhat enough regarding the polymers, although it does not mention previous research that developed smart delivery systems using some of the mentioned polymers (e.g., PAA and Pluronic 123 published by Guang E, Yu B, Xue J. Synthesis of poly(acrylic acid) (PAA) modified Pluronic P123 copolymers for pH-stimulated release of Doxorubicin. Journal of Colloid and Interface Science, 358 (2), 2011), which would strengthen the novelty of this research since it comprises the combination of a more complex system.
Page 6, lines 252-253: The peak at 1.92 ppm doesn’t indicate (d, CH3C-Br) as mentioned in line 253. Check the figure.
What does the peak at 2.1 ppm represent? Does it represent (d, CH3C-Br)?
Page 7, line 282: The author should indicate that the PDI is high so we cannot say that the nanoparticles are monodispersed.
Page 8, lines 309-310: How can you prove that this effect is actually due to a thermo-responsive behavior? There is no thermal analysis (e.g., DSC) that shows the signal corresponding to the volume-phase transition or lower critical solution temperature (LCST) for these systems, considering their composition and concentration.
Page 9, line 321: The sentence “Regarding cell viability studies, Figure (5a)” is not correct. Check it.
Page 9, lines 339-340: How the PDI with a value around 0.4 can indicate the homogeneity and narrow distribution?
Zeta-potential interpretation: The authors should precise the nature of the repulsion leading to stability by interpreting the zeta-potential data measured.
Conclusion section (lines 394-405): As a proof of concept, I believe the study demonstrates physicochemically the pH-responsive behavior (but not completely for the thermo-sensitivity) of the PLCs compared to the Liposomal formulations. However, there's no evidence of the circulation time and whether if opsonization is avoided at a certain extend since in vivo studies are not presented. In addition, it should be relevant to define the target product profile (e.g., regarding the route or administration strategy) for this formulation more than merely mentioning their possible applications for cancer, gene therapy, etc. The previous is also relevant for assessing the particle size of the different formulations in the medium to be used for the administration. Moreover, it should be relevant to provide future perspectives for improving or complementing the research, such as exploring active targeting combined with the multi-stimuli response properties, as well as carrying out in vivo studies.
Reviewer 4 Report
See attached file.

Round 2
Reviewer 2 Report
The authors addressed all the points that I raised so I suggest the pubblication on Pharmaceutics
Author Response
The authors would like to thank the reviewer.
Reviewer 3 Report
The authors have modified the manuscript as suggested but two main points should be clarify.
Answer to Point 2 is still not convenient. Indeed, the smart drug delivery system should also present encapsulation and controlled release properties via internal and/or external stimuli. Therefore, the authors should modify the related sentence.
Answer to Point 12 is off topic. In the Results part the authors claim: “The colloidal stability in water was evaluated by measuring the hydrodynamic diameter and zeta potential of MNP (101 nm, PDI 0.15; −12.3 mV) and MNP-GEM (102.8 nm, PDI 0.174; +15.1 mV).” However, in the Discussion part, there is no relevant information regarding the colloidal stability of their systems. Is a steric, an electrostatic or an electrosteric stability? The authors must propose a discussion to answer to that question.
Reviewer 4 Report
I have accepted the authors's responses.
Author Response

(The authors gave the same response as above.)
